Journal of Machine Learning Research 23 (2024) 1-13        Submitted 6/30; Revised 9/20; Published 10/18

# Benchmarking Embedding Aggregation Methods in Computational Pathology: A Clinical Data Perspective

Shengjia Chen[1,2], Gabriele Campanella[1,2,*], Abdulkadir Elmas[3], Aryeh Stock[4], Jennifer Zeng[4], Alexandros D. Polydorides[4], Adam J. Schoenfeld[5], Kuan-lin Huang[1,3], Jane Houldsworth[4], Chad Vanderbilt[6], and Thomas J. Fuchs[1,2]

[1]Windreich Department of Artificial Intelligence and Human Health, Icahn School of Medicine at Mount Sinai, New York, USA
[2]Hasso Plattner Institute for Digital Health at Mount Sinai, Icahn School of Medicine at Mount Sinai, New York, USA
[3]Department of Genetics and Genomic Sciences, Center for Transformative Disease Modeling, Tisch Cancer Institute, Icahn Institute for Data Science and Genomic Technology, Icahn School of Medicine at Mount Sinai, New York, USA
[4]Department of Pathology, Molecular and Cell-Based Medicine, Icahn School of Medicine at Mount Sinai, New York, USA
[5]Department of Medicine, Memorial Sloan Kettering Cancer Center, New York, USA
[6]Department of Pathology, Memorial Sloan Kettering Cancer Center, New York, USA
*Corresponding author: gabriele.campanella@mssm.edu

**Editor:** My editor

## Abstract

Recent advances in artificial intelligence (AI), in particular self-supervised learning of foundation models (FMs), are revolutionizing medical imaging and computational pathology (CPath). A constant challenge in the analysis of digital Whole Slide Images (WSIs) is the problem of aggregating tens of thousands of tile-level image embeddings to a slide-level representation. Due to the prevalent use of datasets created for genomic research, such as TCGA, for method development, the performance of these techniques on diagnostic slides from clinical practice has been inadequately explored. This study conducts a thorough benchmarking analysis of ten slide-level aggregation techniques across nine clinically relevant tasks, including diagnostic assessment, biomarker classification, and outcome prediction. The results yield following key insights: (1) Embeddings derived from domain-specific (histological images) FMs outperform those from generic ImageNet-based models across aggregation methods. (2) Spatial-aware aggregators enhance the performance significantly when using ImageNet pre-trained models but not when using FMs. (3) No single model excels in all tasks and spatially-aware models do not show general superiority as it would be expected. These findings underscore the need for more adaptable and universally applicable aggregation techniques, guiding future research towards tools that better meet the evolving needs of clinical-AI in pathology. The code used in this work are available at https://github.com/fuchs-lab-public/CPath_SABenchmark

**Keywords:** Computational Pathology, Histopathological Image Analysis, Embedding Aggregation, Benchmark Analysis

## 1 Introduction

Advancements in deep learning have significantly revolutionized the field of computational pathology (CPath), particularly in the analysis of whole slide images (WSIs) (Song et al., 2023; Bilal et al., 2023). Due to their gigapixel resolution, WSIs are usually divided into small tiles for analysis, and weakly supervised learning is a popular training strategy to leverage slide-level supervision without the need of pixel level annotations (Campanella et al., 2018, 2019). Most applications of weakly supervised learning in pathology focus on training slide-level aggregators and using a pre-trained vision model to encode tiles into feature vectors (Campanella et al., 2018, 2019; Lu et al., 2021; Laleh et al., 2022). A notable trend enhancing this capability is the adoption of self-supervised learning (SSL) technique to train foundation models (FMs) on large-scale, domain-specific datasets (Schirris et al., 2022; Campanella et al., 2023; Chen et al., 2023). These models, pre-trained on extensive datasets, provide a robust foundation for task-specific fine-tuning with transfer learning. However, a critical limitation in the field is its reliance on public datasets for downstream task performance evaluation which may not generalize well to a clinical setting. Datasets like TCGA, which, while invaluable for genomic research, may not be ideal for histological analysis. This limitation is not only due to potential biases from high tumor prevalence but also stems from the use of legacy scanning techniques that result in poor cell-level resolution and the reliance on frozen section tissues. These factors collectively contribute to the dataset's limitations, impacting the accuracy and generalizability of histological studies.

**Related Work**  Recent review and benchmark analyses have extensively evaluated AI algorithms' performance and limitations in CPath using WSI datasets. Laleh et al. (2022) applied six weakly supervised algorithms to six clinically significant tasks, driven by a systematic literature review for unbiased selection. Yet, their emphasis on end-to-end pipeline overshadowed the slide-level aggregation phase and lacked exploration of foundation model benefits. Bilal et al. (2023) proposed a comprehensive CPath workflow, assessing seven methods on a single TCGA dataset, which might limit the findings' applicability. Their study aimed at a fair comparison across various aggregation techniques but was constrained by dataset specificity. Saunders et al. (2023) compared five multiple instance learning (MIL) algorithms on three public datasets, highlighting the advantage of ensemble methods for accuracy. However, their study did not engage with clinical-relevant datasets. These studies reveal the need for benchmarking slide-level aggregation methods with a focus on clinically relevant tasks while incorporating embeddings from FMs. They also highlight a gap in discussing the interplay among various method groups and their evolution with artificial intelligence (AI) and computer vision (CV) techniques, essential for enhancing clinical applications.

**Our Contributions**  In this comprehensive benchmarking analysis, we evaluated ten slide aggregation techniques across nine clinically relevant tasks including diagnostic assessment, biomarker classification, and outcome prediction. Our study selected these methods through a structured literature search, prioritizing techniques that advance embedding aggregation technology. Our objectives are summarized as follows: 1. **Benchmarking Widely Used Aggregation Methods**: We prioritized aggregation methods commonly used for comparison in recent work. 2. **Assessment of Embedding from FMs**: To understand the

impact of embedding source on the performance of aggregation methods, we employed embeddings derived from four FMs: three pretrained on domain-specific histological images and the other on the ImageNet dataset. 3. **Providing Insights and Guidelines**: We aim to provide recommendations for effectively using slide aggregation methods. We introduce an evolutionary tree of relevant methods highlighting relationships among them.

## 2 Methods

**Selection of Aggregation Methods:** To select aggregation methods for our study, we conducted a literature search on Google Scholar with the query "((deep learning) AND ((computational histopathology) OR (whole slide images)) AND (classification))" for publications between 2017 and 2023. While recognizing the potential of hierarchical models to fuse local and global features, we focused on embedding aggregation technologies that do not require instance-level labels, drawing from studies that utilize embeddings at a uniform magnification for consistency. Our analysis centers on methods with public, implementable code and those frequently used in recent scholarly work for benchmarking purposes. Problem formulation of benchmarking aggregation methods is provided in Appendix A. Details on the selected aggregation methods and their computational complexities are provided in Appendix C. Table 2.

Figure 1 provides an overview of aggregation methods from 2017 - 2023. In the "Key Instance" category, Campanella et al. (2019) focus on processing a selected subset of the most suspicious tiles sequentially, but not fully considering the spatial relationships between tiles across the entire slide. Ilse et al. (2018) proposed an attention-based aggregation with two fully trainable layers, considering contributions of all instances through attention weights. VarMIL (Schirris et al., 2022; Carmichael et al., 2022) extended attention-based MIL with a variance module for tissue heterogeneity analysis. DS-MIL (Li et al., 2021a) employed a dual stream approach, coupling max-pooling with attention scoring for instance evaluation. Cluster-based approaches like DeepMISL (Yao et al., 2019) and DeepAttnMISL (Yao et al., 2020) integrate phenotype-level information, with Lu et al. (2021) addressing multi-class classification via attention for pseudo label generation. Remix (Yang et al., 2022) reduces WSI bag instances using patch cluster centroids and applies latent space augmentations. These methods focus on instance significance without considering spatial distribution of patches in WSI. AB-MIL (Ilse et al., 2018) and CLAM (Lu et al., 2021) focus on the highest scoring instance with binary and multiclass setting, whereas DS-MIL (Li et al., 2021a) overlooks instance correlations. In contrast, TransMIL (Shao et al., 2021) utilizes self-attention mechanisms within a transformer architecture to analyze spatial information, employing pyramid position encoding. However, its approach to position encoding lacks absolute spatial consistency across WSIs. DT-MIL (Li et al., 2021b) addresses this by incorporating absolute position features alongside a deformable transformer to boost spatial awareness efficiently. Similarly, SET-MIL (Zhao et al., 2022) adopts a token-to-token vision transformer for extracting multi-scale context from WSIs, employing absolute position encoding for precise spatial representation. KAT (Zheng et al., 2022b) further advances this concept by matching tokens with positional kernels. Spatially-aware graph methods enhance image analysis by representing patches as nodes. Patch-GCN (Chen et al., 2021), for instance, constructs graphs from adjacent patches and uses CNNs for effective local-

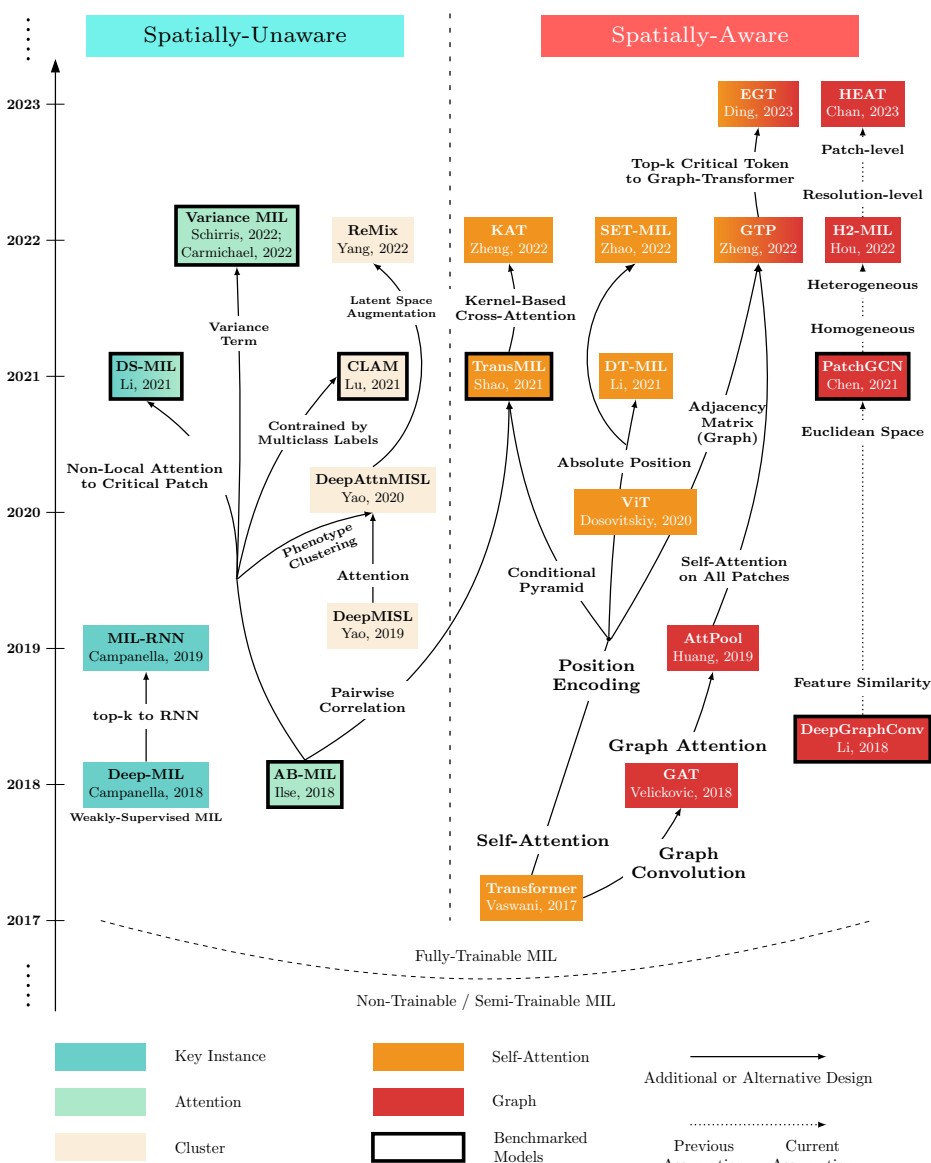

Figure 1: Evolution of Slide Aggregation Methods in CPath (2017 - 2023). We track the progression of aggregation and embedding techniques, categorized by Key Instance, Attention, Cluster, Self-Attention, and Graph-based methods. Models benchmarked in this study are marked with a black outline. Colors and gradient colors denote method categories and their combinations, respectively; vertical placement shows chronological order, and horizontal lines indicate whether spatial information is integrated or not.

to-global information aggregation, outperforming methods like DeepGraphConv (Li et al., 2018) that depend on embedding space similarities. The incorporation of attention mechanisms, introduced by Velickovic et al. (2017), has further advanced this domain. AttPool, applied in CPath, exemplifies this by selecting discriminative nodes for a hierarchical graph and employing attention-weighted pooling for graph representation. GTP (Zheng et al., 2022a) and EGT (Ding et al., 2023) extend these concepts by integrating adjacency matrices for graph construction with self-attention for embedding aggregation on selected top-k critical tokens, respectively. Heterogeneous graph approaches, such as in H2-MIL (Hou et al., 2022), offer innovative strategies for advanced graph representation.

**Large-scale datasets and clinically relevant tasks:** To benchmark aggregation performance on clinical tasks, we collected nine datasets from two institutions, Mount Sinai Health System (MSHS) and Memorial Sloan Kettering Cancer Center (MSKCC). The MSHS slides were scanned on Philips Ultrafast scanners, while the slides from MSKCC were scanned on Leica Aperio AT2 scanners. The cohorts included are described below and summarized in Appendix B. Table 1. Histogram of number of tiles per slides can be found in Appendix C. Figure 3B.

**1. BCa**: Breast cancer (BCa) detection cohort. Breast cancer blocks and normal breast blocks were obtained from the pathology laboratory information system. A total of 1998 slides were sampled, with 999 positive and 999 negative. The positive slides were selected from blocks that received the routine biomarker panel for cancer cases (estrogen receptor: ER, progesterone receptor: PR, human epidermal growth factor receptor 2: HER2, and Ki67), while negative slides were selected from breast cases that did not have an order for the routine panel. Additionally, negative cases were selected if they were not mastectomy cases, did not have a synoptic report associated with the case, and had no mention of cancer or carcinoma in the report. **2. IBD:** Inflammatory Bowel Disease (IBD) detection cohort. Normal mucosa samples were obtained from patients undergoing screening and routine surveillance lower endoscopy from 2018 to 2022. IBD cases, including first diagnoses and follow-ups, were included. A total of 1441 slides were sampled, 717 with active inflammation and 724 with normal mucosa. **3. BCa ER, BCa PR, BCa HER2:** BCa biomarker prediction cohorts. Breast cancer cases are routinely assessed for ER, PR, and HER2 status using immunohistochemistry (IHC) and Fluorescence In Situ Hybridization (FISH). Results for each biomarker were automatically extracted from the pathology report. **4. BioMe BR HRD:**: Breast (BR) Homologous Repair Deficiency (HRD) prediction cohort. Mount Sinai BioMe is a whole-exome sequencing cohort of 30k individuals, where carriers of pathogenic and protein-truncating variants affecting HRD genes, i.e., BRCA1 BRCA2 BRIP1 PALB2 RAD51 RAD51C RAD51D ATM ATR CHEK1 CHEK2, where included as positives. A subset of the BioMe dataset of patients with available breast pathology slides were included. Slides containing solely normal breast tissue and slides with breast cancer were both included. **5. LUAD EGFR:** EGFR (Epidermal Growth Factor Receptor) mutation status prediction in Lung Adenocarcinoma (LUAD). Two datasets were collected, one from MSHS and one from MSKCC. For the MSHS dataset, a total of 294 slides were obtained from MSHS clinical slide database, 103 positive and 191 negative. The cohort was built following the guidelines described in previous work Campanella et al. (2022) to map mutations to a binary target. The MSKCC dataset consists of 1,000 slides with 307 posi-

tive and 693 negative and is a random subset of the dataset described in Campanella et al. (2022). **6. NSCLC IO:** Lung cancer immunotherapy response prediction. Non-small cell lung cancer (NSCLC) patients who received PD-L1 blockade-based immunotherapy were considered. Cytology specimens were excluded. The objective overall response was determined by RECIST Eisenhauer et al. (2009) and performed by a blinded thoracic radiologist. A total of 454 slides were obtained, 86 positive and 368 negative.

**Model Implementation Details:** We used the same experimental setup for all datasets and tasks. The embeddings as the input of the aggregation module were generated from four pretrained FMs: 1. Truncated ResNet50 (**tres50_imagenet**, dim:1024), pretrained on ImageNet (Lu et al., 2021). 2. **CTransPath** (dim:768) integrates a CNN and Swin Transformer, pretrained on 5.6 million tiles from TCGA and PAIP datasets (Wang et al., 2022). 3. DINO-ViT small (**dinosmall**, dim:384), pretrained on 1.6 billion histological images from over 420,000 clinical slides (Campanella et al., 2023). 4. **UNI** (dim:1024) leverages ViT-L/16 via DINOv2, pretrained on over 100 million images from 100,000 WSIs from public and in-house datasets (Mass-100K) (Chen et al., 2023). Each dataset was split using a Monte Carlo Cross-Validation (MCCV) strategy where for each MCCV split, 80% of the samples were used for training and the rest for validation. We generated 21 MCCV folds for each task; one was used exclusively for hyperparameter tuning, and the subsequent 20 folds were each run twice using fixed random seeds (0 and 2024) to assess model variance and robustness. All models were trained with a single A100 GPU for 40 epochs using AdamW Loshchilov and Hutter (2017) optimizer. A cosine decay with a 10-epoch warm up schedule was used for the learning rate and weight decay hyperparameters. Implementation details for each method follow the instruction on official sources but keep consistent training strategy across all experiments. A GPU memory usage of each methods to number of tiles per slide is summarized in Appendix C. Figure 3A.

## 3 Results

We assessed aggregation methods using the area under the receiver operating characteristic curve (AUC) averaged over 20 MCCV runs. The AUC is reported at the end of training (40 epochs). Boxplots were utilized to compare AUC distributions for embeddings generated by four FMs, presented in same subfigures of Figure 2. Overall, domain-specific embeddings (CTransPath, dinosmall, UNI) outperform tres50_imagenet across most tasks, regardless of the aggregator used. Additionally, using UNI generally yields better performance in lung datasets. We observe that no method is consistently superior to the others across all datasets and embeddings. To compare different methods within the same embedding, one-sided t-tests were conducted to determine whether each method is significantly better than the AB-MIL as baseline. In BCa and BCa PR detection, no consistent advantage of advanced methods over baseline approaches is observed, except transMIL and PatchGCN showed a statistically significant improvement over the baseline ($p < 0.05$) with CTransPath input. In BCa ER and IBD detection tasks, several methods are superior to the baseline AB-MIL for all four embeddings as input. PatchGCN and AB-MIL_FC exhibit strong results in MSK LUAD EGFR tasks. In the MS LUAD EGFR tasks, all methods perform on par with or less effectively than the baseline, with the dataset's smaller size (294 slides) potentially influencing these outcomes. In BCa HER2 and BIOME HRD tasks, no significant differences

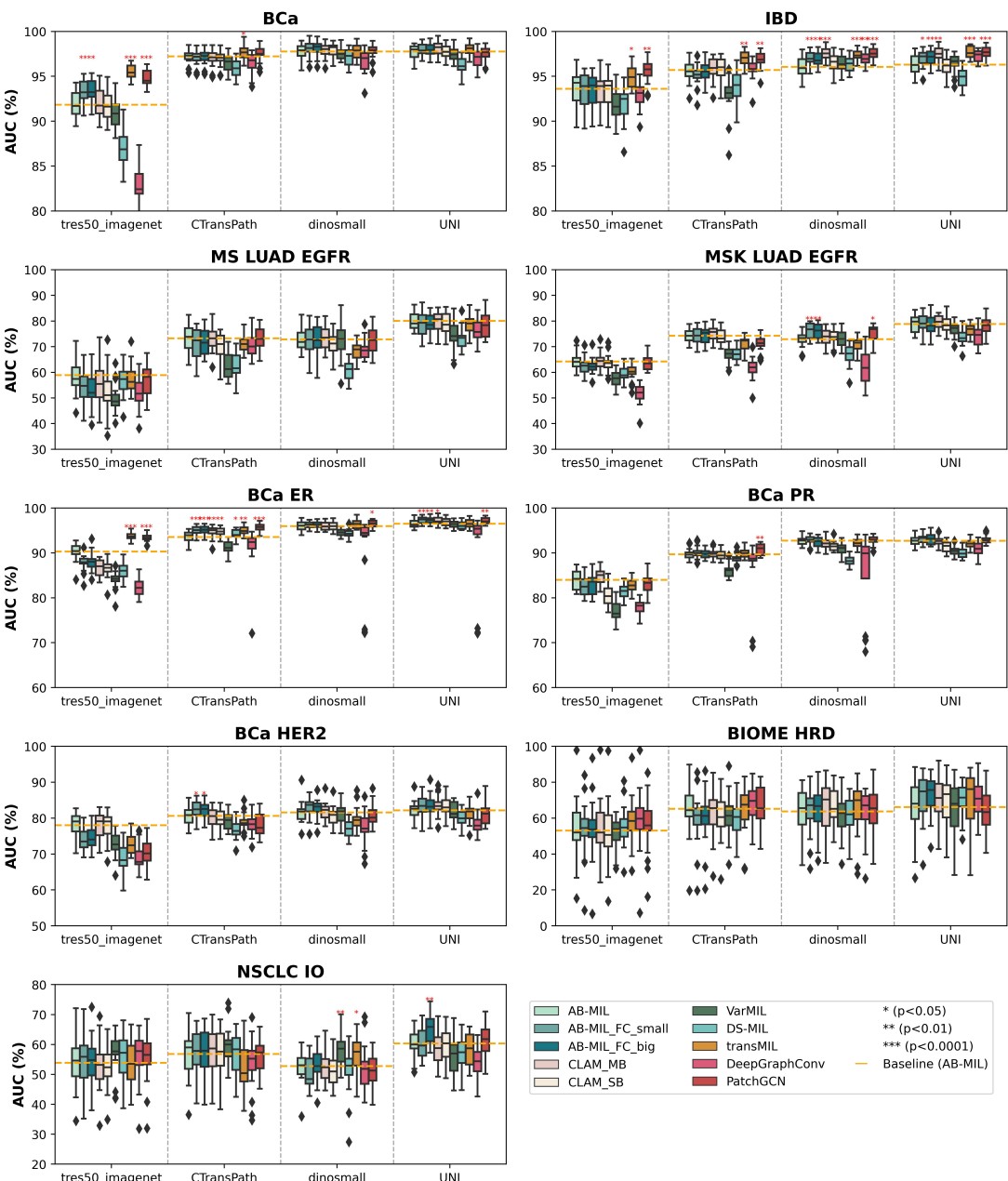

Figure 2: AUC scores in boxplots from benchmark aggregation methods versus AB-MIL baseline across nine datasets, using four embedding groups. Scores are from 20 Monte Carlo cross-validations, averaged over two random seeds. A one-sided t-test assessed AB-MIL performance comparisons, with symbols indicating significant differences. The dotted orange line shows the AB-MIL average for reference. Methods follow the Figure 1 category order and colors.

are observed across all embeddings and aggregators. However, domain-specific embeddings exhibit less variance in the box plots. In outcome prediction for NSCLC IO, VarMIL and transMIL show superiority with dinosmall input, while AB-MIL_FC_big performs better with UNI; however, the improvements are marginal and exhibit high variability. To show the speed of convergence, an example curve of validation AUCs during training process is in Appendix C. Figure 3C.

## 4 Discussion

In this study, we employed ten aggregation methods, utilizing embeddings from ImageNet and several domain-specific FMs, to benchmark performance across nine clinically relevant datasets. Our empirical evaluation reveals that: (1) for general disease detection tasks, attention mechanisms (AB-MIL) are effective, albeit additional spatial information could be incorporated at a computational cost; (2) using transfer learning directly from FM pretrained on natural images to histological images without domain-specific tuning may degrade results, yet proficient aggregation methods can diminish this performance gap; (3) for more challenging tasks such as outcome prediction or replicative biomarker prediction, the inclusion of spatial information using current methods contributes marginally to performance. Based on these findings, while it is clear that pathology FMs provide superior performance, it is not possible to recommend any particular aggregation method. We suggest using AB-MIL as a strong baseline and validate other methods on a case by case basis.

Despite the growing number of aggregation algorithms published, there is no clear evidence for a method that is consistently superior than AB-MIL. The inclusion of spatial information, while theoretically sound, has yet to yield the expected gains. It is possible that spatial information is not relevant for certain tasks, but this is in contradiction with pathologists' intuition. More likely, current methods fall short and better ways to incorporate spatial information across slides are needed. Future research should focus on developing methods that can better leverage spatial context and hierarchical structures designed for WSIs. In future work, we will expand our datasets to include a wider range of clinically relevant tasks, including survival analysis. We are developing infrastructure for secure benchmarking of external models on our clinical cohorts, which we plan to share with the community. We will explore the most recent FMs for more nuanced representations and investigate class activation map visualizations to enhance the interpretability and effectiveness of aggregation methods.

## Acknowledgments and Disclosure of Funding

This work is supported in part through the use of research platform AI-Ready Mount Sinai (AIR.MS) and the expertise provided by the team at the Hasso Plattner Institute for Digital Health at Mount Sinai (HPI.MS). This work was supported in part through the computational and data resources and staff expertise provided by Scientific Computing and Data at the Icahn School of Medicine at Mount Sinai and supported by the Clinical and Translational Science Awards (CTSA) grant UL1TR004419 from the National Center for Advancing Translational Sciences.

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

## Appendix A. Problem Formulation

Let a WSI be denoted by $X$, representing a 'bag' in the MIL framework. The bag $X$ comprises a set of instances $\{x_1, x_2, \ldots, x_N\}$, where each instance $x_i$ is a tile extracted from $X$. An encoder function $f(\cdot)$ transforms each instance $x_i$ into a low-dimensional embedding, resulting in a set of embeddings $\{h_1, h_2, \ldots, h_N\}$, where $h_i = f(x_i)$ represents the feature vector for the $i$-th tile. Optionally, spatial information $s_i$ can be associated with each $x_i$, capturing its position within $X$. The aggregation function $g(\cdot)$ aggregates the set of all embeddings and optionally their spatial information to output a single vector $H$ that serves as the bag-level representation for the entire WSI $X$. This can be expressed as: $H = g(\{(h_1, s_1), (h_2, s_2), \ldots, (h_N, s_N)\})$. The final prediction $Y$ for the bag $X$ is then obtained by applying a suitable classifier $c(\cdot)$ to $H$: $Y = c(H)$.

## Appendix B. Summary of Datasets

Table 1: Summary of benchmark datasets in this study. BCa: Breast Cancer, IBD: Inflammatory Bowel Disease, ER: Estrogen Receptor, EGFR: Epidermal Growth Factor Receptor, LUAD: Lung Adenocarcinoma, IO: Immunotherapy Response, NSCLC: Non-small cell lung cancer.

| Code | Origin | Task | Disease | Slides | Tiles [min, max] |
|------|--------|------|---------|--------|------------------|
| BCa | MSHS | Disease Detection | Breast Cancer | +999,-999 | [22, 40086] |
| IBD | MSHS | Disease Detection | IBD | +717, -724 | [159, 16009] |
| BCa ER | MSHS | Replicative Biomarker | Breast Cancer | +1000,-1000 | [291, 30564] |
| BCa HER2 | MSHS | Replicative Biomarker | Breast Cancer | +1258, -760 | [291, 34946] |
| BCa PR | MSHS | Replicative Biomarker | Breast Cancer | +1033, -953 | [291, 30564] |
| BIOME BR HRD | MSHS | Replicative Biomarker | Breast Cancer | +375, -188 | [69, 37849] |
| MS LUAD EGFR | MSHS | Replicative Biomarker | LUAD | +103, -191 | [61, 45339] |
| MSK LUAD EGFR | MSKCC | Replicative Biomarker | LUAD | +307, -693 | [18, 44128] |
| NSCLC IO | MSKCC | Outcome Prediction | Lung Cancer | +86, -368 | [13, 44128] |

## Appendix C. Summary of Selected Methods and Model Parameters

Table 2: A comparison of selected aggregation methods in terms of size, and number of parameters.

| Abbreviation | Authors | Group | Size (MB) | # Params (M) |
|---|---|---|---|---|
| AB-MIL | Ilse et al. 2018 | Attention | 4.51 | 1.182 |
| DeepGraphConv | Li et al. 2018 | Graph | 3.01 | 0.789 |
| DS-MIL | Li et al. 2021 | Attention | 0.59 | 0.153 |
| TransMIL | Shao et al. 2021 | Self-Attention | 10.19 | 2.671 |
| CLAM | Lu et al. 2021 | Attention | 3.02 | 0.790 |
| Patch-GCN | Chen et al. 2021 | Graph | 5.27 | 1.38 |
| VarMIL | Schirris et al. 2022; Carmichael et al. 2022 | Attention | 2.02 | 0.529 |

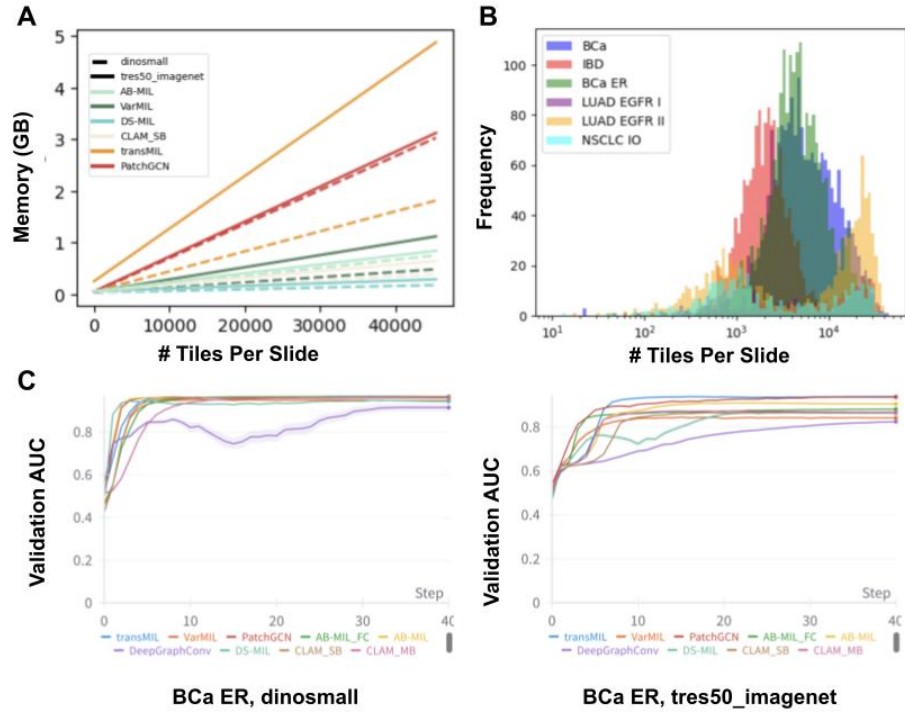

Figure 3: A: Computational resources vs. tiles per slide; B: Histogram of number of tiles per slide in each dataset; C: Validation AUC during training process for BCa ER. The line is average value of validation AUC and errorbar is calculated by standard error from 20 MCCV runs.

