# OpenReview forum: "Benchmarking Embedding Aggregation Methods in Computational Pathology: A Clinical Data Perspective"
_MICCAI.org/2024/Workshop/COMPAYL — COMPAYL 2024_

### Official Review · Reviewer_j7vk · 2024-07-08
**Review of Paper 6**

**Custom Rating:** 5
**Confidence:** 5

**Review:**

This paper conducts extensive research and systematic benchmarking on the encoding stages (ResNet50, CTransPath, DINO, UNI) as well as aggregation strategies (MIL, GCN, CLAM, and up to 10 different methods) across multiple WSI-level prediction task scenarios.
This is an exciting piece of work, and its findings make us to rethink the increasingly complex WSI-level prediction pipelines. I am eagerly looking forward to the Github repository of the code and related data or embeddings (if possible) to become publicly available resources that facilitate the comparison of different methods.

The references on the first two lines of page 6 ([3]) are not updated. The case style of the xlabel in Figure 2 should be consistent with the main text. Everything else is perfect!

---

### Official Review · Reviewer_c68z · 2024-07-08
**Review of Benchmarking Embedding Aggregation Methods in Computational Pathology**

**Custom Rating:** 4
**Confidence:** 4

**Review:**

The paper conducts a study comparing the performance of different aggregation schemes for multiple-instance learning across different clinical tasks and feature extractors. The results suggest that while domain-specific foundational models (FM) are important, space-aware methods (ie using graphs) do not significantly improve performance and it is not possible recommend any particular aggregation method. The authors suggest continuing with AB-MIL as baseline.

Pros:
- The paper addresses a significant topic in the field of computational pathology (CP) and the usage of FM for slide-level applications.
- The text is well written and it is easy to follow.
- A wide variety of datasets is used.
- The authors carry on statistical test to verify the significance on the difference in performance between methods.

Cons/comments:
- While ROC-AUC is usually employed for evaluating binary classifiers, its behavior on imbalanced datasets is not intuitive, as it can return good results if the model is biased towards the majority class. Some of the datasets used in your study present some imbalance, specifically NSCLC IO for Outcome Prediction. I'd suggest using another metric such as PR-AUC or F1-Score. Some of the conclusions drawn for this dataset may be wrong.

Overall I think it's a good paper, it addresses an open problem in CP and obtains interesting, although counterintuitive, conclusions (as the authors mention, we'd expect spatial-aware methods to outperform structure-less methods).

---

### Official Review · Reviewer_xDLj · 2024-07-15
**A useful experimental review on the value of multiple slide-level aggregators across several clinical tasks**

**Custom Rating:** 5
**Confidence:** 5

**Review:**

**Overview**
This paper analyzes a set of ten different ways of aggregating tile-level features into a slide-level feature used for slide-level classification.
Starting with a literature search on PubMed, the authors identified aggregation methods based on some inclusion criteria (based on publication date, absence of instance-level labels required, focus on a single magnification level, etc.) and applied them to a set of clinically relevant tasks using cohorts from private sources (MSKCC and MSHS). Figure 1 in the paper gives a nicely detailed overview of methods presented since 2017 and how they have evolved over the years.
As summarised by the authors, no single approach stands out in terms of performance, AB-MIL is confirmed to be a strong baseline method to use across cohorts, and the use of spatial information can add some benefits but has not been used optimally, yet.
The results emerging from this study may give useful information to the CPATH community, although these results are only applicable to the cohorts used in this paper and it's unclear how well this can generalise to other tasks or datasets.
Overall, this work highlights the need for benchmarks based on applications that are very close to clinical practice, based on fairly large and non-public datasets (given the large use that the community has done of datasets such as TCGA), but that should be made accessible to the public to allow others to benchmark their models as well, something that the authors mention in the discussion as work that they are currently doing in this direction.

**Pros**
* Pretty comprehensive analysis of several aggregation methods
* Used cohorts are fairly large
* Code is available
* Results and discussion points, based on the authors' experience on this work are useful for the computational pathology community
* All experiments can be run on a single GPU

**Cons**
* Data is not publicly available
* Inclusion criteria might be relaxed to include other methods in the future
* Cross validation was done, unclear how these models will perform on data from different sources; given the presence of data from two centres, it would have been interesting to test the robustness of models trained on data from one center to data from another center. However, since pre-trained encoders were used, the question about robustness would apply to the aggregation method only.

---

### Decision · Program_Chairs · 2024-07-16

Accept